# Gait Training in Virtual Reality: Short-Term Effects of Different Virtual Manipulation Techniques in Parkinson’s Disease

**DOI:** 10.3390/cells8050419

**Published:** 2019-05-06

**Authors:** Omar Janeh, Odette Fründt, Beate Schönwald, Alessandro Gulberti, Carsten Buhmann, Christian Gerloff, Frank Steinicke, Monika Pötter-Nerger

**Affiliations:** 1Human Computer Interaction, Department of Informatics, University of Hamburg, D-22527 Hamburg, Germany; janeh@informatik.uni-hamburg.de (O.J.); frank.steinicke@uni-hamburg.de (F.S.); 2Department of Neurology, University Medical Center Hamburg-Eppendorf, 20215 Hamburg, Germany; o.fruendt@uke.de (O.F.); b.schoenwald@uke.de (B.S.); a.gulberti@uke.de (A.G.); buhmann@uke.de (C.B.); gerloff@uke.de (C.G.); 3Department of Neurophysiology and Pathophysiology, University Medical Center Hamburg-Eppendorf, 20215 Hamburg, Germany

**Keywords:** Parkinson’s disease, gait training, virtual reality, rehabilitation, gait asymmetry, freezing of gait

## Abstract

It is well documented that there is a strong relationship between gait asymmetry and the freezing of gait (FOG) in Parkinson’s Disease. The purpose of this pilot study was to find a “virtual reality (VR)- based” gait manipulation strategy to improve gait symmetry by equalizing step length. Fifteen male PD patients (mean age of 67.6 years) with FOG were assessed on a GAITRite^®^ walkway. Natural gait was compared with walking conditions during “VR-based” gait modulation tasks that aimed at equalizing gait symmetry using visual or proprioceptive signals. Compared to natural gait, VR manipulation tasks significantly increased step width and swing time variability for both body sides. Within the VR conditions, only the task with “proprioceptive-visual dissociation” by artificial backward shifting of the foot improved spatial asymmetry significantly with comparable step lengths of both sides. Specific, hypothesis-driven VR tasks represent an efficient tool to manipulate gait features as gait symmetry in PD potentially preventing FOG. This pilot study offers promising “VR-based” approaches for rehabilitative training strategies to achieve gait symmetry and prevent FOG.

## 1. Introduction

Gait disturbances represent the main symptom impacting every-day self-dependence and quality of life in Parkinson’s disease (PD) patients. Gait disturbances and freezing of gait (FOG) promote frequent falls occurring in up to 87% of PD patients [1,2,3], which result in hospitalization, immobilization, and a loss of self-confidence. The Parkinsonian gait disturbance is characterized by continuous features [4] as reduced speed, shorter step length, increased stride-to-stride variability, reduced automaticity, and increased gait asymmetry [5,6] as well as episodic phenomena such as freezing episodes, festination, and starting arrests [4]. The freezing episodes are closely associated with continuous gait characteristics particular to the degree of gait asymmetry [7,8]. With regard to limited therapeutic options of medication or deep brain surgery, physical training strategies have evolved to be a focus of interest to improve gait and the freezing of gait.

Virtual reality (VR) has emerged as an efficient tool in physical rehabilitation [9,10] in PD. VR offers the opportunity to simulate immersive, controllable, changeable environments with the option to create individualized, specific training programs. To achieve a more natural experience and highly immersive VR, simulations are often generated using a head-mounted display (HMD) such as HTC VIVE^®^. This system represents a low-cost, easy accessible, portable pedestrian simulator system, which can be used in different settings, in the lab, at home, or during physiotherapy. “VR-based” interventions attempt to promote neuroplasticity and motor learning [11,12,13]. Motor learning strategies (MLS) consider specific motor learning principles, which are “patient” and “task-specific.” VR offers the opportunity to facilitate the incorporation of motor learning principles such as real-time multisensory feedback, task variation, objective progression, and task-oriented repetitive training [9]. VR has been shown to improve balance and gait especially in PD [9,10,14], particularly when combined with conventional rehabilitation. However, to date, many of the VR studies lacked a clear rationale for intervention programs and did not utilize motor learning principles. Particularly VR training strategies developed by theory-driven protocols may assist motor learning implementation for optimized “VR-based” treatments. Implementation of patient-tailored motor learning strategies into the design and planning of VR interventions may enhance the efficiency and improve the therapeutic outcome [9].

Recently, novel insights revealed the heterogeneity of the freezing of gait in PD with distinct freezing phenotypes as asymmetric-motor, anxious, and sensory-attention phenotypes [15]. Particularly, the strong relationship between gait asymmetry and freezing of gait in PD patients [7,8] represents a therapeutic clue. Reduced episodes of FOG might be attained by establishing enhanced lower limb gait symmetry. Dopaminergic medication seems to promote improved gait symmetry in PD patients [8]. Deep brain stimulation (DBS) of the subthalamic nucleus improves gait symmetry and freezing of gait by reducing the amplitude of the better side (“better side down”) [16], which can be used for trouble-shooting in PD patients with FOG and DBS. Specific physiotherapeutic approaches designed to achieve gait symmetry [5] or using treadmill training [7] or split belt-locomotion [17] effectively reduced gait asymmetry and FOG.

The rational of the current study is to apply the equalization of gait asymmetry as the motor learning principle by use of VR techniques with real-time multisensory feedback to increase efficacy of the training strategy. Specifically, we aim to exploit and optimize the VR environment by variation of different conditions to define the method with the best possible equalization of the pathological gait asymmetry in view of step length. This would lead to a theory-driven, individualized therapeutic approach, which might be used in future studies for long-term gait symmetry trainings to reduce FOG and falls in PD patients.

## 2. Materials and Methods

The study was conducted in accordance with the Declaration of Helsinki (World Medical Association, General Assembly, Helsinki, Finland, June 1964) and approved by the local Ethics Committee of the Medical Council in Hamburg and the local Ethics Commission of the Department of Informatics in Hamburg (reference number PV5281). All participants gave written informed consent.

### 2.1. Subjects, Clinical Data, and Questionnaires

A total of 16 male patients with idiopathic PD participated in the study. They were recruited using an announcement from the outpatient clinic/Parkinson day clinic of the Department of Neurology of the University Medical Center Hamburg-Eppendorf. Inclusion criteria were diagnosis of idiopathic PD, according to the UK PD Society Brain Bank criteria [18] and the criteria of the German Society of Neurology (Deutsche Gesellschaft für Neurologie, DGN), Hoehn & Yahr stage 2–3 [19], no deep brain stimulation or medication pump, clinically relevant gait disorder with freezing, but without the tendency to fall, walking independently without a walking aid, and normal vision, if necessary, with a vision aid. Other inclusion criteria include no severe dementia (Montreal Cognitive Assessment MoCA [20] > 21, this cut-off has also been used in Reference [21]) as dementia can be associated with gait disturbances [22], Giladi’s Freezing of Gait Questionnaire [23] with a score > 6 (this cut-off has been used in [24]), no severe polyneuropathy (pallaesthesia > 4/8), and no spinal stenosis or severe musculoskeletal disorders that impair sensorimotor function or gait.

All patients were tested in the “on” medication state, which was defined as about 1 h after the intake of their regular medication. Prior to the main experiment, patients were characterized by a short interview, demographic questions, and clinical scores and questionnaires.
Unified Parkinson Disease Rating Scale of the Movement Disorder Society (MDS-UPDRS) part III [25] as a general motor scoreGerman version of the freezing of gait questionnaire by Giladi [23,26] as a subjective assessment of FOG in PDZiegler’s freezing of gait course [27] as an objective assessment of FOG in PDShort, 7-item version of the Berg balance scale [28] as an objective measure of balance as a parameter of gait stabilityGerman version of the Montreal cognitive assessment (MoCA, [20], http://www.mocatest.org) as a measure of cognitive function in PDGerman version of the Parkinson’s Disease Questionnaire (PDQ-39, [29,30]) as a measure of quality of life in PDSimulator Sickness Questionnaire (SSQ [31]) before and after the experimentSlater, Usoh, and Steed Questionnaire (SUS [32]) as a measure of presence in the virtual environment

One patient withdrew from the study due to exhaustion while 15 male patients with idiopathic PD completed the experiment. Table 1 gives an overview of patient characteristics as well as results of clinical scores and questionnaires.

### 2.2. GAITRite^®^ and Virtual Reality (VR)

Gait parameters were measured using a GAITRite^®^ electronic walkway system (https://www.gaitrite.com/ [33]). The GAITRite^®^ consists of a walkway with overall dimensions of 90 cm × 7 m × 3.2 mm. The virtual space was rendered using Unity3D (https://unity3d.com/de/unity) and showed an outdoor scene with a long grass pathway (see Figure 1b). We included a virtual mat in the virtual environment (VE) that exactly matched the real GAITRite^®^ system walkway. The virtual item correlates to the start (green line) and target (red line) lines were placed on the floor in front of the participant to indicate the walking distance in the virtual world as well as the real world. We used an Intel computer for rendering with 2.7 GHz Core i7-6820HK processor, 16GB of main memory and Nvidia GeForce GTX 1070 graphics cards. The participants wore an HTC Vive HMD device for the stimulus presentation manufactured by Vive (https://www.vive.com/de/product/). In order to provide a realistic VR scenario while walking, we attached the Vive controllers to the participant’s leg to present virtual feet in VR resembling the participant’s real feet. We tracked sensors on the HMD (i.e., that tracked the user’s head movement) and the Vive controllers (i.e., that tracked the user’s leg movement) using a Lighthouse tracking system with sub-millimeter precision in the laboratory.

### 2.3. Experimental Procedure

We performed the experiment in a laboratory room of 9 m × 4 m in size (see Figure 1a), which was shielded from light and noise. After providing the patients with the full equipment, participants were asked to first assume the start position by standing in an orthostatic pose at the start line. Then, participants were instructed to walk at their self-selected pace along the walkway of the GAITRite^®^ system while stopping at the location of the target line (see Figure 1b). After each trial, the participant had to walk back to the starting point alongside the GAITRite^®^ walkway. During the experiment, we provided comfort to the head from the weight of the cables by having an assistant manage the cables for each participant. The total time per participant, including questionnaires, interview, instructions, experiment, individual breaks, and debriefing was about 1.5 to 2 h.

### 2.4. Gait Modulation Conditions

Seven different walking conditions were performed with three repetitions each (trials 1, 2, and 3) and a duration of about 5 to 6 min. After a short familiarization phase to become used to the setup of each trial, patients were asked to walk at their self-selected pace and start using the dominant leg, which was labeled as the leg that was predominantly used in the pull test to stabilize the stance.

Gait analysis was divided into two parts:Conditions without gait asymmetry equalization as a motor learning strategy (non-MLS): consisting of three walking conditions 1. Natural walk, 2. Walk with diving glasses to detect the influence of field of view (FOV), and 3. Walk with HTC Vive without specific gait asymmetry manipulation in VR.Conditions with gait asymmetry equalization as a motor learning strategy (MLS): including 4 walking conditions to assess the effects of different VR gait manipulation strategies using visual targets and proprioceptive signals on gait asymmetry and other gait parameters (see Table 2).

In the end, a natural walk was repeated to ensure gait stability over time and explore possible after-effects of short-term application of VR gait manipulation tasks. The hypothesis-driven, different conditions are explained in detail in Table 2. For a better understanding, exemplary step lengths of one patient are given to clarify the mode of VR gait manipulation.

### 2.5. Gait Parameters

For all seven conditions mentioned above, spatio-temporal gait parameters were analyzed through the GAITRite^®^ walkway system. The mean value of all the three trials per condition was used to calculate the individual gait parameters. For the non-MLS and MLS conditions, we focused on gait variability and gait asymmetry. The following gait parameters were used (for further details on gait parameters, see the Gait Rite Manual (https://www.procarebv.nl/wp-content/uploads/2017/01/Technische-aspecten-GAITrite-Walkway-System.pdf): Step length (cm), velocity (cm/s), cadence (steps/min), the gait asymmetry index (G_i_Asym corresponding to the ratio of longer step length/short step length adapted from References [17,34]), stride velocity (cm/s), step time (s), step width (cm), double support time (s), and swing time (s). As a measure of gait variability, we used the coefficient of variation (CV) of the previously mentioned gait parameters. The CV was already introduced in previous studies [35,36] (%CV = [(Standard deviation/Mean value) × 100]. Furthermore, we used the leg length of each participant (see Table 1) to calculate a functional ambulation performance (FAP) score derived by subtracting points from a maximum score of 100 for walking at a self-selected velocity [37]. A higher score is better in overall walking performance.

### 2.6. Data Analysis

We confirmed the assumptions of the ANOVA for the experimental data. The results were normally distributed according to a Shapiro-Wilk test at the 5% level. Degrees of freedom were corrected using Greenhouse and Geisser estimates of sphericity in case the assumption of sphericity had been violated. Data analyses were performed using the IBM SPSS software version 25.0.

For exploratory reasons, we first performed an overall ANOVA to look for gait velocity differences between each of the three trials (trials 1, 2, and 3) during all nine conditions to check if the self-selected pace of the patients differed between the trials or conditions.

Afterward, for non-lateralized gait parameters (velocity, cadence, gait asymmetry, and FAP score), we performed repeated measures one-way ANOVAs to evaluate the effects of the different gait manipulation methods on these parameters (three factors in the “non-MLS conditions” [“baseline”, “diving glasses,” and “real virtual”], 5 factors in the “MLS conditions” [“baseline”,“symmetrical without feet”, “symmetrical with feet”, “asymmetrical with feet”, “visual-proprioceptive dissociation”]. For lateralized gait parameters, we analyzed the results with two-way repeated-measures ANOVA (gait modulation method × body side). For each condition, the term “body side” represented the leg with longer or shorter step length during the baseline condition. For the testing of after-effects, the natural gait condition at the beginning and at the end of the experimental session were compared. Post-hoc effects were calculated at the 5% significance level. We have also provided partial eta-squared (η_p_^2^) effect sizes to supplement the interpretation of the results. The interpretation of η_p_^2^ suggests that effect sizes greater than 0.01 but less than 0.06 be considered small, greater than 0.06 but less than 0.14 be considered medium, and greater than 0.14 be considered large [38].

The SSQ questionnaire scores before and after the experiment were compared using the non-parametric Wilcoxon Signed Rank Test at the 5% significance level. Spearman’s correlation coefficient was used to correlate clinical data (age, Ziegler score, Berg balance score, MoCA, and MDS-UPDRS III, SUS) with the gait parameters mentioned above in the baseline condition. Thus, 17 parameters were used for correlations, which needed the Bonferroni correction for multiple testing (p Bonferroni = 0.003).

## 3. Results

### 3.1. General Aspects

In six patients, the left side was the most affected body side (side with symptom onset and ongoing symptom predominance) whereas the other eight patients had pronounced symptoms on their right body side. In only five out of fifteen patients (= 33.3%), the clinically most affected body side represented the leg with the shorter step length. None of our PD patients represented FOG during the experiment even though all patients claimed freezing in the questionnaires, which is a common phenomenon in experimental settings [39]. Nine patients used their left leg and six patients used their right leg as the dominant leg in the pull test (= starter leg). Gait parameters did not differ between conditions in which PD patients started walking with the dominant or non-dominant leg. Although patients were asked to walk at their self-selected pace and performed a training session before each recorded condition, gait velocity during the first trial of each condition was significantly slower compared to the second and third trial, which might be explained by a prolonged adaptation to the changing conditions (see Figure 2 including Appendix A).

### 3.2. Objective Gait Parameters in Virtual Reality (VR)

Objective gait parameters of PD patients in the different “non-MLS conditions” (“baseline,” “diving glasses,” “real virtual”) and “MLS conditions” (“symmetrical without feet,” “symmetrical with feet,” “asymmetrical with feet,” and “manipulated”) are shown in Figure 2.

#### 3.2.1. Conditions Without Asymmetry Equalization as Motor Learning Strategy (non-MLS): Effects of Field of View and “Pure” Virtual Environment

For detailed results of the “non-MLS conditions,” see Appendix A. The two-factorial ANOVA of lateralized gait parameters with the intra-subject factors gait manipulation method (“baseline,” “diving mask,” and “real virtual”) × body side (short side, long side) revealed a significant main effect of the factor “gait manipulation method” on step time (F_1.76, 24.68_ = 4.93, *p* = 0.01, η_p_^2^ = 0.261), step width (F_1.62, 22.7_ = 13.48, *p* < 0.001, η_p_^2^ = 0.491), swing time (F_1.63, 22.92_ = 9.15, *p* = 0.002, η_p_^2^ = 0.395), and step time variability (F_1.9, 26.61_ = 3.97, *p* = 0.03, η_p_^2^ = 0.221). The factor “body side” was significant for step length (F_1, 14_ = 42.62, *p* < 0.001, η_p_^2^ = 0.753), step time (F_1, 14_ = 7.64, *p* = 0.01, η_p_^2^ = 0.353), and swing time variability (F_1, 14_ = 10.49, *p* = 0.006, η_p_^2^ = 0.428). One-way ANOVA revealed a significant main effect of gait modulation on cadence (F_1.8, 25.91_ = 7.39, *p* = 0.003, η_p_^2^ = 0.346). Post-hoc tests for both ANOVAs with significant differences of gait parameters within the different gait conditions are shown in Appendix A and Figure 2a).

In post-hoc tests, there was no significant difference between the “baseline” condition and “diving glass” condition with regard to all gait parameters. These findings indicate that reducing the FOV to the size we used in the experiment by use of HMD did not impact gait (see Figure 2a).

Furthermore, while walking with a head-mounted display (HMD) in the “real-virtual” condition, the step length was comparable to the baseline condition, but other gait parameters changed significantly. PD participants walked with a decreased step time, an increased step time variability, a decreased swing time, a widening step width, and increased cadence (see Figure 2a and Appendix A) indicating a rather insecure gait pattern in VR.

#### 3.2.2. Conditions With Asymmetry Equalization as Motor Learning Strategy (MLS): Effects of Different VR Gait Manipulation Conditions on Gait

The two-factorial ANOVA with the intra-subject factors “gait manipulation method” (“baseline,” “symmetrical without feet,” “symmetrical with feet,” “asymmetrical with feet,” “manipulated”) × body side (short side/long side) revealed a significant effect of gait manipulation method on step length (F_2.21, 30.99_ = 8.85, *p* = 0.001, η_p_^2^ = 0.387), step time (F_2.65, 37.17_ = 3.67, *p* = 0.02, η_p_^2^ = 0.208), swing time (F_2.43, 34.1_ = 3.39, *p* = 0.03, η_p2_ = 0.195), step width (F_3.05, 42.74_ = 10.8, *p* < 0.001, η_p_^2^ = 0.436), step time variability (F_2.97, 41.57_ = 11.51, *p* < 0.001, η_p_^2^ = 0.451), and swing time variability (F_1.95, 27.43_ = 6.18, *p* = 0.006, η_p_^2^ = 0.306). The factor “body side” revealed significant differences only for step length (F_1, 14_ = 6.63, *p* = 0.02, η_p_^2^ = 0.322). One-way ANOVA revealed a significant main effect of gait modulations only on cadence (F_3, 42.07_ = 5.44, *p* = 0.003, η_p_^2^ = 0.280). Furthermore, a repeated measure ANOVA determined that the mean asymmetry index (G_i_Asym) differed statistically and significantly between gait manipulation methods (F_2.82, 39.54_ = 3.17, *p* = 0.037, η_p_^2^ = 0.185). Post-hoc tests of gait parameters within the different gait manipulation conditions are shown in Appendix A and Figure 2b). There were no significant interaction effects between the gait manipulation method × body side for all gait parameters.

Post-hoc tests revealed that the different gait manipulation strategies have been associated with an increase in the step width, step time variability, and swing time variability for both sides compared to the baseline condition reflecting an insecure gait. Asymmetry of gait (G_i_Asym) significantly decreased only during a “manipulated” virtual foot condition with visual-proprioceptive dissociation. Step length of the short side approached the step length of the long side, which reflects a lower degree of gait asymmetry in the “manipulated” condition compared to the baseline condition (F_2.82, 39.54_ = 3.17, *p* = 0.037, η_p_^2^ = 0.185). In summary, the manipulated condition with visual-proprioceptive dissociation was the most effective method to reduce gait asymmetry and to adjust step lengths of both legs.

#### 3.2.3. After-Effects of VR Gait Modulation Conditions on Gait and Gait Asymmetry

Comparison of natural gait at the beginning and at the end of the experiment revealed no significant differences for any gait features. 2-factorial ANOVA revealed no significant effects for the factor gait modulation method or body side for all gait parameters. Post-hoc tests with differences for gait parameters are shown in Appendix A.

### 3.3. Simulator Sickness and Presence

SSQ scores of PD patients before and after the experiment are given in Table 1. They indicate low simulator sickness symptoms for walking with an HMD, and we found no significant increase of symptoms over the time of the experiment (Z = −1.02, *p* = 0.306). The mean SUS Questionnaire scores for the sense of feeling present in the virtual environment was M = 3.5 SD = 0.8, which indicates a moderate sense of presence. Additionally, participants judged their fear to collide with the walls of the room or other physical obstacles while immersed with the HMD during the experiment as comparably low (i.e., along a 0–5-point rating scale, 0 = no fear, 5 = high fear, M = 0.38 SD = 0.47).

### 3.4. Correlations

Clinical characteristics of PD patients as age, cognition (MoCA), motor score (MDS-UPDRS III), gait and freezing related scores (Berg balance score, Ziegler’s FOG score), and the stimulation sickness scores (SUS, pre-/post-SSQ) were not significantly correlated with any of the gait parameters after the Bonferroni correction for multiple testing.

## 4. Discussion

The main purpose of this project was to develop a theory-driven, individualized therapeutic rehabilitative approach by defining optimized motor learning strategies implemented in VR for the best possible equalization of the pathological gait asymmetry in PD. To our knowledge, this is the first study with the specific “VR-based” approach to implement gait symmetry as the motor learning principle in PD.

The present study revealed three main findings, which are of relevance in the design of future rehabilitative VR training strategies. First, during a natural walk in the baseline condition, PD patients revealed the typical, clinically relevant gait asymmetry with a significant difference between step lengths of both legs. The leg with the shorter step length was only in 33.3% of patients the clinically the most affected one contradicting the clinical impression that the range of motion decreased more on the clinically most affected side. This finding implies that, in future gait symmetry trainings, the first step should be to define the “worst and best side” by measuring step lengths in quantitative gait analyses. Second, walking in VR induced an increase of step width, cadence, and gait variability, which indicates an insecure gait pattern during the VR conditions. This is not due to a simple reduction of the field of view, since walking with diving glasses did not impact the gait pattern. Similarly, in a previous study, we found alterations of nearly all tested gait parameters in healthy subjects using HMD [40,41]. Third, we found the optimized equalization of gait asymmetry in the VR condition with dissociation of the visual and proprioceptive inputs (“manipulated” condition). PD patients overcame the spatial asymmetry and exhibited a comparable step length by enlarging the step length of the short side, an adapted step time, and a swing time variability of both sides in this “manipulated” condition. Thus, our findings suggest that virtual walking with visual-proprioceptive dissociation might have important implications for the restoration of gait symmetry in PD patients with FOG [5,7,17].

Virtual reality provides a unique platform to study the complex interactions between an individual’s movement and the environment. VR has tremendous potential to advance both our understanding and treatment of gait impairments. It has been applied successfully in PD patients in previous studies for motor and cognitive assessment, motor trainings, and rehabilitation [9,10]. In a currently published review, evidence for a positive effect of the VR exercise on certain gait parameters as step and stride length has been found with overall effects on gait, balance, and quality of life that were comparable to that of physiotherapy [10]. It has been found to be a safe method without relevant adverse effects [42], which is easily accessible for private use as well. In addition, the use of VR increases motivation and enjoyment that may assist motor learning by ensuring continuous training.

In view of the current hypothesis of gait asymmetry in the pathophysiology of FOG and falls, there have been rehabilitative approaches focused on the improvement of gait asymmetry using split-belt treadmill training [17], conventional physiotherapy [5], or treadmill training with visual cues on a screen and acoustic feedback [7]. Those training strategies on gait symmetry revealed beneficial effects. However, they were partly unspecific [7] or induced even intermittent, short-term deterioration with the “better side-down” strategy [17]. The advantage of this specific manipulation of gait symmetry in the current study is the exploitation of immersive VR techniques such as proprioceptive-visual dissociation, which is not available outside the VR. It remains to be elucidated how specific and effective this training strategy is in future rehabilitative training sessions.

The superiority of visual-proprioceptive dissociation in VR compared to other VR conditions was quite astonishing. Most of the current rehabilitative training strategies such as the Lee Silverman Voice Treatment (LVST BIG) therapy focus on the improvement of self-awareness and recalibration of movements [43]. The hypothesis of the current rehabilitative motor learning strategies is that basal ganglia sensorimotor processing deficits result in disturbed amplitude scaling and bradykinesia in PD [44]. Partially, PD patients might overcome this sensorimotor deficit by focusing on movement execution counteracting the under-scaling of the amplitude. However, when the conscious focus on movement execution stops, PD patients fall back into the automated gait pattern with restricted, under-scaled movements. LSVT-BIG therapy aims at the conscious recalibration of the under-scaled movement amplitudes with continuous, graded implementation of the newly gained movement pattern into increasingly complex everyday activities to transport the recalibrated movement template into automated behaviour [43]. In our specific, effective VR paradigm, we “deceived” the patient’s self-awareness of the movement by visual-proprioceptive dissociation instead of reinforcement of self-awareness.

Recently, utilization of error–driven motor learning has been implemented in gait training strategies [45,46]. In post-stroke patients, gait asymmetry due to hemiparesis was artificially exaggerated on the split-belt treadmill by augmentation of the velocity differences of both sides [45,46]. The hypothesis was that error augmentation is necessary to drive the nervous system to make corrections. Error augmentation induced an adjusted gait pattern with short-term and long-term effects after repeated training sessions in stroke patients [46]. In our specific, effective VR paradigm, we deceived the patient´s self-awareness of the movement by visual backward shifting of the shorter leg resulting in error augmentation. This might be the reason for the effectiveness of that particular VR manipulation method. Due to current positive long-term findings of error augmentation in stroke patients after repeated training sessions [46], we are confident that the “VR-based” multi-sensory error augmentation might also be effective in the long-term in PD patients.

In the “manipulated” VR condition, the manipulated foot induces a conflict between the visual and proprioceptive signals about foot position. Such a conflict can be used in different ways to estimate the way in which visual and proprioceptive signals are synchronized. The present study shows that our PD subjects mapped their real foot movements onto corresponding movements of the manipulated virtual foot in the virtual world. Therefore, the consistency can be derived from the responses to the adjustments between proprioceptive and visual information. PD patients initially showed notable asymmetry that was gradually adjusted toward symmetry during the “manipulated foot” condition.

In another proprioceptive study, the “rubber hand illusion” was assessed in PD [47]. In this experimental paradigm, synchronous stroking of a rubber hand and the subject’s hidden real hand resulted in the illusory experience of ‘feeling’ the rubber hand and proprioceptive mis-localization of the real hand toward the rubber hand (‘proprioceptive drift’). PD patients predicted larger proprioceptive drift as compared to healthy controls. The amount of the proprioceptive drift was correlated with disease duration and interpreted as deficient multisensory integration in cortico-basal ganglia-thalamic circuitry in PD [47]. Nevertheless, although PD seems to affect illusory perceptions of body ownership, it was found that dopaminergic treatment appears to increase suggestibility generally rather than having a specific effect on own-body illusions [47]. As in our study, all patients were in the medication “on” state and results demonstrated sufficient gait symmetry in the “manipulated” condition. We assume that suggestibility in the sense of mapping the visually perceived foot onto the own body was present in our PD sample. To our knowledge, there is no data on identical tests performed in healthy subjects, but similar setups suggest that the manipulation of visual-proprioceptive integration in virtual environments lead to gait adaption in healthy subjects as well [48].

Study limitations were a small PD patient sample size with predominantly male participants and the lack of control groups such as PD patients without freezing or age-matched healthy controls. Furthermore, information on the body mass index should consistently be recorded since obesity can also alter gait patterns [49]. Since we currently only assessed short-term effects, we need to determine the effects of long-term training in the future. Current investigations are underway in our clinic to examine whether training using the “manipulated” condition over a few weeks can lead to long-term improvement of step length symmetry following virtual walking. Although we used a very realistic setup with over-ground walking and spatial consistency of the real and virtual world, an even more realistic and “dynamic” setup might be helpful in the future such as the “moving environment” or using wireless or transparent glasses.

The transfer of motor learning principles attained during the training sessions to everyday life is key for the development of new rehabilitation strategies. Particularly in PD, the transfer of exercised movements in the training session into the daily routine is often unsuccessful and constitutes a serious limitation of rehabilitative approaches [50]. It remains to be assessed whether the use of proprioceptive-visual dissociation in VR is a rehabilitative training strategy that overcomes these potential drawbacks.

## 5. Conclusions

This study presents the results of different “VR-based” gait manipulation methods in PD patients with FOG with the main purpose to find a method equalizing step length asymmetry in PD patients. We found a significant step length difference between both legs in PD with FOG. The virtual dissociation of visual and proprioceptive signals was most promising in accomplishing this goal and might therefore be a sufficient rehabilitative technique to achieve gait symmetry and, hypothetically, to prevent FOG. Future studies are needed to further investigate the long-term training effects of this specific visual-proprioceptive VR manipulation technique.

## Figures and Tables

**Figure 1 cells-08-00419-f001:**
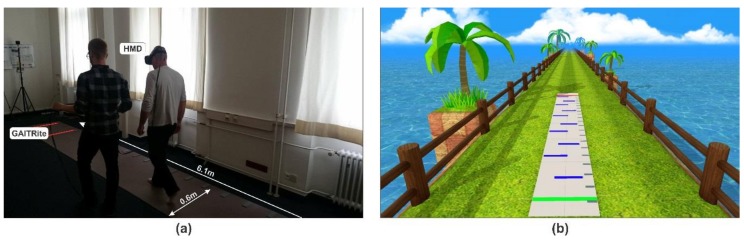
Experimental setup: (**a**) A participant walks in the real workspace with a head-mounted display (HMD) over the GAITRite^®^ walking surface. (**b**) Participant’s view of the virtual environment on the HMD. The walkway in the virtual environment exactly matched the real GAITRite^®^ system walkway.

**Figure 2 cells-08-00419-f002:**
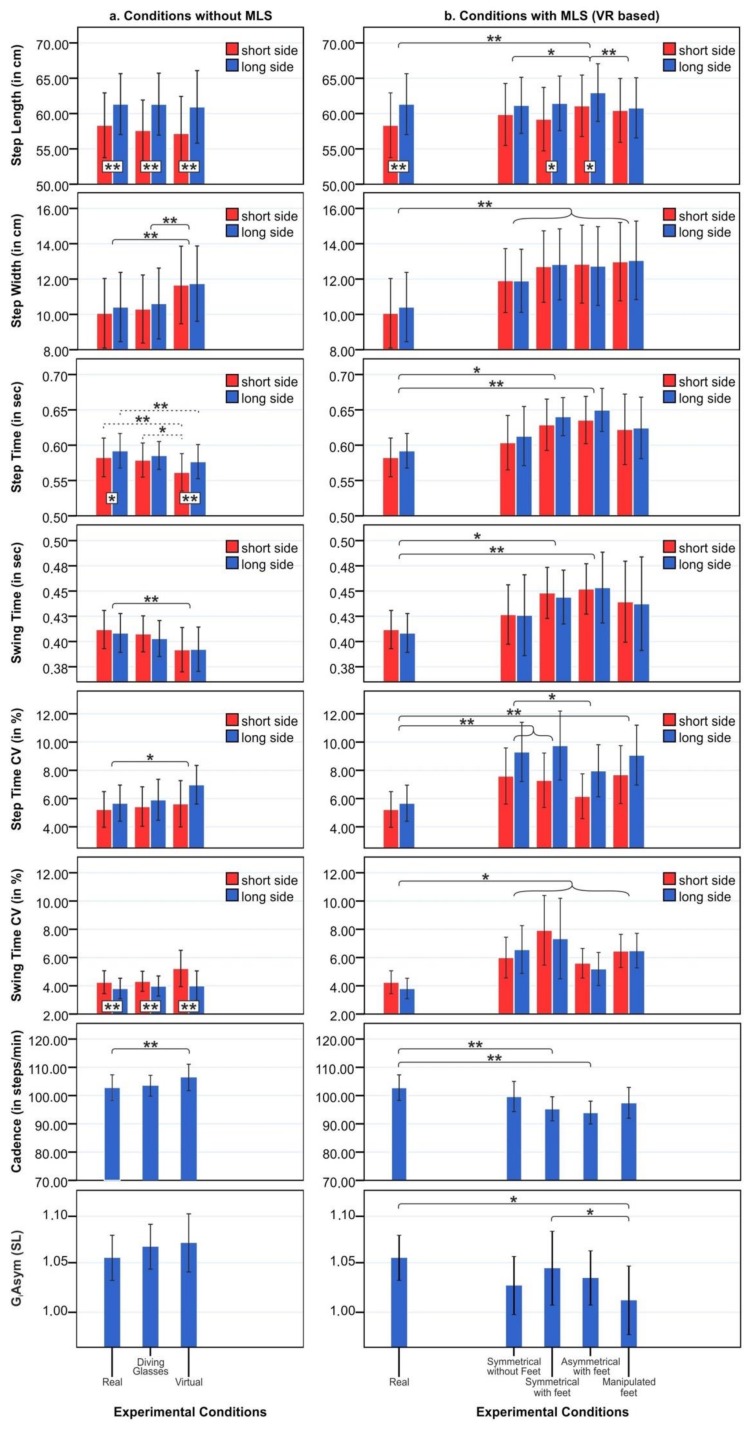
General gait parameters for (**a**) “Conditions without motor learning strategies (non-MLS)” and (**b**) “Conditions with MLS (“VR-based”, MLS)” are given as mean values +/- standard error of mean. Significant differences are marked as follows: (*) = *p* < 0.05, (**) = *p*< 0.01 and (***) = *p* < 0.001. 
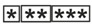
 = significant difference between short and long side 
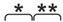
 = significant difference between different gait modulations conditions for short and long sides 
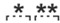
 = significant difference between gait modulations conditions for either the short or the long side.

**Table 1 cells-08-00419-t001:** Patient characteristics, clinical data, and questionnaires.

Patient Characteristics and Clinical Data	Mean ± SD (min–max)
**Age**	67.6 years ± 7 (49–77)
**Gender**	15 male
**Handedness**	1 ambidextrous, 14 right handed
**Hoehn & Yahr Scale**	2–3
**Levodopa (Minutes After Intake)**	58.3 minutes ± 19.8 (40–120)
**Leg Length**	Left: 93.8cm ± 3.7 (88–102cm) Right: 93.7cm ± 4.0 (88–102cm) Leg with shorter step length: 93.7 cm ± 4.0 (89–102 cm) Leg with longer step length: 93.8 cm ± 3.8 (88–102 cm)
**Onset of PD Symptoms**	11.5 years ± 4.9 (2–19)
**PD Diagnosis**	9.5 years ± 4.9 (1–17)
**Onset of Gait Disturbance**	5.5 years ± 4.4 (1–17)
**MDS-UPDRS part III**	25.5 ± 7.2 (12–37)
**Giladi’s FOG**	27.5 ± 10.6 (12–47)
**Berg Balance**	24.7 ± 1.8 (19–26)
**Ziegler’s FOG**	7.2 ± 5.9 (0–17)
**MoCA**	27.5 ± 2.0 (23–31)
**Pre-SSQ**	16.45 ± 16.59 (3.74–52.36)
**Post-SSQ**	15.21 ± 17.04 (3.74–56.1)
**SUS**	3.5 ± 0.8 (1–5.83)
**PDQ-39**	25.31 ± 12.83 (5.76–45.21)

MDS-UPDRS part III: motor part of Unified Parkinson Disease Rating Scale of the Movement Disorder Society. Giladi’s FOG: Giladi Freezing of Gait questionnaire as a subjective measure of FOG. Ziegler’s FOG: Objective assessment of the Freezing of Gait. MoCA: Montreal cognitive assessment. Pre-SSQ: Simulator Sickness Questionnaire before the experiment. Post-SSQ: Simulator Sickness Questionnaire after the experiment. SUS: Slater, Usoh, and Steed Questionnaire to capture the subjects’ impression of the presence in VE. PDQ-39: Parkinson’s Disease Questionnaire to evaluate quality of life.

**Table 2 cells-08-00419-t002:** Overview of experimental conditions and hypotheses.

Condition	Specification	Hypothesis and Purpose	Example (Step Length)	Illustration
**A. Non-MLS conditions**				
(1) Real World Natural Walk (“Baseline“)	Walking naturally on the GAITRite^®^ pad without glasses * * The glasses were reversed and positioned on the participants’ head to ensure the same weight and posture during each condition	Baseline measurement	Step length of the longer side: 60 cm Step length of the shorter side: 54 cm	-
(2) Real World Natural Walk with Diving Glasses (“Diving Glasses”)	Walking naturally on the GAITRite^®^ pad with diving glasses * * The diving glasses had a similar weight and field of view compared to the HTC Vive^®^	To detect a possible impact of a peripheral limitation of the participants’ field of view on gait, e.g., gait instability or slowing down of gait.	Step length of the longer side: 60 cm * Step length of the shorter side: 54 cm * * optimally, gait parameters should not be affected by the diving glasses	
(3) Natural Walk in Virtual Reality without visual targets (“Real Virtual”)	Walking naturally on the GAITRite^®^ pad with HTC Vive 3D glasses presenting the virtual environment without visual targets	To detect possible influences of the virtual environment on gait, e.g., gait stability	Step length of the longer side: 60 cm * Step length of the shorter side: 54 cm * * Optimally, gait parameters should not be affected by the 3D glasses	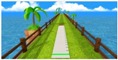
**B. MLS conditions**				
(4) Walking in Virtual Reality with symmetric setup without presenting the feet(“Symmetrical without feet”)	Walking on the GAITRite^®^ pad with HTC Vive 3D glasses. Lines are presented each with a distance (d) that corresponds to the individuals’ step length of the longer side - there are no feet presented on the screen. Distance = step length of the longer side – step length of the shorter side (d= SLl – SLs) New step length = old step length + distance (SLs_new = SLs_old + d)	Participants are asked to step on the lines, but walk at the natural speed. To evaluate if the visual target signals might lead to greater gait symmetry.	Step length of the longer side: 60 cm * Step length of the shorter side: 60 cm * * optimally, step length of the shorter side should adapt to that of the longer leg	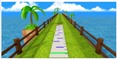
(5) Walking in Virtual Reality with a symmetric setup while presenting the feet (“Symmetrical with feet”)	Walking on the GAITRite^®^ pad with HTC Vive 3D glasses. Lines are presented each with a distance that corresponds to the individuals’ step length of the longer leg - two feet are presented on the screen. Distance = step length of the longer side – step length of the shorter side (d = SLl − SLs) New step length = old step length + distance (SLs_new = SLs_old + d)	Participants are asked to step on the lines with the middle of their feet, but walk as normal as possible while remaining in the middle of the pad. To evaluate if multiple visual signals (target and proprioceptive signals) might influence gait symmetry.	Step length of the longer side: 60 cm * Step length of the shorter side: 60 cm * * optimally, step length of the shorter leg should adapt to that of the longer leg	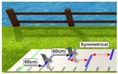
(6) Walking in Virtual Reality with an asymmetric setup while presenting the feet (“Asymmetrical with feet”)	Walking on the GAITRite^®^ pad with HTC Vive 3D glasses. Lines are presented with asymmetrical distances: Step length of shorter leg (SLs) is exaggerated: New step length = step length of the shorter side + (2* (step length of the longer side – step length of the shorter side) (SLs_new = SLs+ (2*(SLI-SLs))) 2 feet are presented on the screen moving kinematically similar to the participants feet	Participants are asked to step on the lines with the middle of their feet, but walk as normal as possible while remaining in the middle of the pad. To evaluate if an exaggeration of step lengths of the shorter leg is needed to achieve gait symmetry.	New step length of the shorter side: 66 cm Step length of the longer side: 60 cm * * optimally, step length of the shorter leg should adapt to that of the longer leg	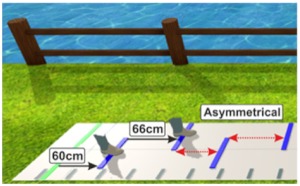
(7) Walking in Virtual Reality with a symmetric setup while manipulating the feet of the shorter side (“visual-proprioceptive dissociation”)	Walking on the GAITRite^®^ pad with HTC Vive 3D glasses. Lines are presented with a distance that corresponds to the individuals’ step length of the longer leg. Two feet are presented on the screen. However, the foot on the shorter side is visually shifted backwards. Shift of the manipulated virtual foot = step length of the longer side – step length of the shorter side (Shift_m = SLI-SLs)	To evaluate if a visual shifting of the proprioceptive signal leads to a greater gait symmetry.	Distance between lines: 60 cm Shift: 6 cm Step length of the longer side: 60 cm * Step length of the shorter side: 60 cm * * Optimally, step length of the shorter side should adapt to that of the longer side.	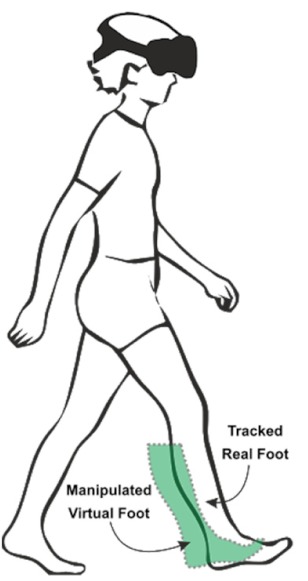

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
