# Peer review of "Gait Training in Virtual Reality: Short-Term Effects of Different Virtual Manipulation Techniques in Parkinson’s Disease"

_cells, 2019, doi:10.3390/cells8050419_

Round 1

Reviewer 1 Report

REVIEW for “Gait training in virtual reality: evaluation of different virtual manipulation techniques in Parkinson’s Disease”.

This present pilot study explores different VR-based gait manipulation strategies that can improve gait symmetry by equalizing step length. The study is original, well designed and sound. The manuscript is well written with a high overall merit. However, the authors should address some minor points that would improve the overall quality of this study.

GENERAL COMMENTS

Introduction: Why do you utilize HMD? It is well documented that VR environments can alter spatiotemporal characteristics during gait. However, there is a small number of studies that highlight the negative effects of HMD due to increased motion-sickness, lack of visual and spatial awareness etc. The current trend for walking in immersive environments is the CAVE systems & see-through HMDs and not the typical binocular HMDs.

Is this a convenient, inexpensive, portable……option? You don’t refer to HMD in the introduction.

How is your approach/interventino better than those described in references 5,7 and 14?

What’s your rationale for using these specific motor learning strategies?

Methods: Did you control for speed OR did you ask your subjects to walk at their self-selected pace? As shown in your supplementary table, there were no differences in walking speed among conditions. Did subjects perform 3 trials at various speeds and you average them out. What if there were large differences among trials?

Did you ask your subjects to walk at the same pace as the “baseline” during the HMD or at whatever pace was comfortable for them? Speed should be compromised due to the weight of the HMD, as well as the reduced spatial awareness and somatosensory information (and maybe simulation sickness).

Results: Increased variability in temporal parameters is associated with unstable gait pattern or instability. However, increased step width is rather a result of insecurity and not instability. Someone that has a wider step width is actually more stable.

Regarding your “discussion” section, I feel that you are just presenting your findings without getting into the underlying mechanisms for these findings (only 1 paragraph). You should have a better integration of the existing literature (you can also include studies in stroke with error augmentation; see Reisman 2009 & 2013) and your current findings. You should also list the limitations of your study.

You are reporting two surprising and quite intriguing findings: a) the leg with the shorter steps was only in 33.3% of patients the one that was clinically most affected, and b) the improvement of leg asymmetry due to the visual-proprioceptive dissociation condition. As you stated, most of the current rehabilitative strategies such as LVST BIG therapy focus on the improvement of self-awareness and recalibration of movements. Based on this, you will have expected the “asymmetrical with feet” condition that promotes exaggeration of the short leg to have more positive outcomes than the “manipulated feet” condition. Please, comment on this.

SPECIFIC COMMENTS

Title & Abstract:
 You may want to consider changing your title to reflect the major finding of your study (optional)

OR add “short term”, “acute effect” to the existing title

Abstract: use “VR-based” (hyphenated). You should correct this throughout the document.

First sentence can be written: “It is well documented that there is a strong relationship between gait asymmetry and freezing of gait (FOG) in Parkinson’s Disease”

Introduction:
The Introduction should be brief and need to contain no more than five paragraphs. Focus on the primary question you are addressing. You need to pose the questions and provide the rationale by citing past literature. The final paragraph of the Introduction should explicitly and specifically state what hypotheses you intend to explore.

Page 1 last sentence needs rewording.

Page 2, first paragraph: add “as” to the first sentence “Virtual reality (VR) has emerged as an efficient……”

Page 2, first paragraph: task specific should be hyphenated

Page 2, second paragraph: “Deep brain stimulation…..trouble-shooting option.” Consider rewording

Page 2, third paragraph: use acronym for “motor learning strategies”. Add “in Parkinson’s Disease” at the end of the first sentence.

Page 2, third paragraph: correct to “We aim specifically i) to assess……..ii) to ascertain”

Methods:
Page 4, 2.2: add the manufacturer for GAITRite and HTC Vive HMD

Table 2: you should spell out SLl & SLs or any acronym that you are using

Page 8, 2.5: spell out FAP

Results:

Page 8, 3.1, last line: Nine patients instead of “9”

Page 10, Legend for Figure 2: correct “moto” to “motor”

Page 10, 3.2.1.: paragraphing

Page 10, 3.2.1.: omit “slightly” when referring to unstable gait pattern (it’s also insecure)

Page 10: be consistent with your F values (subscripts?)

Page 11, top paragraph: step length instead of “step size”

Discussion:

Page 12: paragraphing

Page 12, fifth paragraph: you may want to spell out LVST

Page 12, third paragraph: omit “slightly”, correct “instable” to “unstable”
Page 12, last paragraph: “deceived” instead of “betrayed”

Page 13, first paragraph, first sentence: omit “previous”

Page 13, first paragraph, last sentence: consider rewording (unclear)

Author Response

See Word File below

Reviewer 2 Report

The authors aimed to find a virtual reality (VR)-based gait manipulation strategy to improve gait symmetry by equalizing step length. The authors report that VR manipulation tasks significantly increased step width and swing time variability for both body sides. In particular, the task with “proprioceptive-visual dissociation” by artificial backward shifting of the foot improved significantly spatial asymmetry with comparable step lengths of both sides.

Comments

1.         Were patients' body weight and body mass index taken into account? Was there any influence of body weight and/or obesity on the gait of PD patients that could condition the results described herein?

2.         The number of enrolled and tested patients is relatively small.

3.         Age-matched normal persons were not analyzed in the present study. Is there information, if any, on results of similar/identical tests performed on normal subjects?

4.         The scene "seen" by the PD patients in the VR tests is static. Given that PD patients live in "dynamic" environments (i.e. characterized by moving persons and/or objects), one wonders if the tests described herein wouldn't have provided more complete information using a "dynamic" scene.

5.         The authors should indicate the limitations of their study.

Author Response

We thank the reviewers for the helpful comments and productive critique to the manuscript that we would like to address in the following document (see sections in italic). Please find attached the new version of our manuscript including the changes you suggested marked in yellow color.

REVIEWER 2

Comments

1.        Were patients' body weight and body mass index taken into account? Was there any influence of body weight and/or obesity on the gait of PD patients that could condition the results described herein?

We agree that this is an important aspect and included this in our limitation section (see page 14, last paragraph). Unfortunately, we did not consistently gathered information on the patients’ body weight or BMI. Nevertheless, we measured leg lengths for both body sides for each patient which is important to calculate different gait rite parameters (please see https://www.procarebv.nl/wp-content/uploads/2017/01/Technische-aspecten-GAITrite-Walkway-System.pdf for further information). The results are now included in our Table 1.

Notwithstanding, we will pay attention to the connection of BMI and gait for future studies as it has previously been shown that there are alterations in gait pattern in patients with obesity (Meng et al.,2017).  

2.         The number of enrolled and tested patients is relatively small.

We definitely agree and included this aspect in our newly incorporated “limitation” section at the end of the discussion (page 14, next to last paragraph).

3.         Age-matched normal persons were not analyzed in the present study. Is there information, if any, on results of similar/identical tests performed on normal subjects?

Thank you for this important note. We now included the missing control group into our limitations section (page 14, next to last paragraph).

Nevertheless, we do not expect healthy subjects to present relevant body side differences regarding step length, which is why we intentionally did not include healthy subjects.

To our knowledge, there is no data on identical tests performed in healthy subjects, but similar set-ups suggest that the manipulation of visual-proprioceptive integration in virtual environments lead to gait adaption in healthy subjects as well (Cano Porras et al., 2017).

4.         The scene "seen" by the PD patients in the VR tests is static. Given that PD patients live in "dynamic" environments (i.e. characterized by moving persons and/or objects), one wonders if the tests described herein wouldn't have provided more complete information using a "dynamic" scene.

Indeed, a more dynamic environment (e.g. moving objects) might give a more realistic information on gait in PD. However, moving objects could also stand for more distraction in the sense of dual tasking and would therefore represent another confounder. To make patients focus on our specific task aiming at the equalization of step length to possibly reduce freezing of gait, we decided to choose a static environment to eliminate any other confounders that might influence gait parameters (as e.g. dual-tasking is known to induce freezing; see Spildooren et al., 2010).

Nevertheless, we agree, that future studies should increase the cognitive load e.g. by using dynamic environments to create a more realistic atmosphere.

5.         The authors should indicate the limitations of their study.

We now included a limitation section at the end of the discussion in our manuscript (see page 14, next to last paragraph).

Round 2

Reviewer 2 Report

The authors have satisfied my previous criticism.